# The Pharmacists of Physical Activity: Physiotherapists Empowering Older Adults’ Autonomy in the Self-Management of Aging with and Without Persistent Conditions

**DOI:** 10.3390/healthcare13070834

**Published:** 2025-04-06

**Authors:** Mike Studer, Kent Edward Irwin, Mariana Wingood

**Affiliations:** 1Department of Physical Therapy, Touro University, Las Vegas, NV 89014, USA; 2Department of Physical Therapy, University of Nevada, Las Vegas, NV 89154, USA; 3Physical Therapy Program, Midwestern University, Downers Grove, IL 60515, USA; 4Department of Implementation Science, Wake Forest University School of Medicine, Winston-Salem, NC 27101, USA; 5Department of Internal Medicine, Section on Gerontology and Geriatric Medicine, Wake Forest University School of Medicine, Winston-Salem, NC 27101, USA

**Keywords:** physical activity, aging, frailty, behavior change, exercise, physical therapy, physiotherapy

## Abstract

Aging has been thought to be factual, inherited, and obligatory. However, aging can be divided into primary (i.e., inevitable physiological changes) and secondary (i.e., age-associated changes driven by life choices, environment, and society) aging. The impact of social norms and life choices is why no two 70-year-olds look the same. The life choice that appears to have the strongest impact on aging is physical activity. Research continues to highlight the power of mitigating age-related losses via physical activity and debunking the notion that age-related changes such as falls, frailty, and functional decline are inevitable. Physiotherapists are the healthcare professionals who reverse or slow down age-related changes and prevent secondary aging from occurring. Physiotherapists are the health profession’s experts in movement science, whose interventions primarily center around physical activity as medicine. Thus, physiotherapists function as pharmacists of physical activity and are well-positioned to prescribe the dosages needed for wellness promotion as well as disease prevention and management. This paper provides guidance from the perspective of the physiotherapist on exercise prescription most optimal and consumable for an older population.

## 1. Introduction

Aging has been thought to be factual, inherited, and obligatory; however, it can be divided into primary (i.e., inevitable physiological changes) and secondary (i.e., age-associated changes driven by life choices, environment, and society) aging [1,2]. Direct international healthcare costs for people not meeting the recommended levels of physical activity have been estimated to reach $520 billion by 2030 [3]. A majority of these costs are attributed to chronic diseases and secondary aging that could be prevented, managed, or reversed by following one of the suggested levels of physical activity offered as a guideline by the World Health Organization (WHO) in 2009 [4]. The consensus statement, which operates as a guideline, recommends that people of all ages participate in 200–300 min of physical activity each week. Alternatively, the guidelines make an important distinction that intensity is healthier. Individuals who are not as risk averse should perform 75–150 min of high intensity interval training. The WHO Physical Activity guidelines are woven into this paper, yet the reader should note the intentional inclusion of the words “physical activity” and, to a lesser extent, “exercise”. Physical activity is defined as any movement that requires energy expenditure that is greater than what is used at rest [5]. Physical activity includes active transportation, recreational activities, housework, yardwork, and exercise [6]. Longitudinal and randomized control intervention studies continue to illustrate that falls, frailty, and functional decline can be prevented, managed, or reversed through physical activity [6,7,8,9].

Due to the benefits of physical activity, including exercise, it is often referred to as medicine. In fact, Dr. Rober Butler (the founding director of the National Institute of Aging) is frequently quoted as saying, “If exercise could be packaged in a pill, it would be the single most widely prescribed and beneficial medicine in the nation.” Despite the abundance of physical activity research, aging, falls, frailty, and dependence on others are thought to be factual, inherited, and obligatory components of aging. This perception results in a loss of an internal locus of control or a reduction in self-efficacy. Correlated and in some cases causal with this loss, individuals are less likely to be physically active, feeling that the effects of aging are out of their control. These perceptions are related to stereotype embodiment, stereotype threat, and ageism, which result in a self-fulfilling prophecy of age stereotype, such as not exercising or being physically active [10]. In fact, studies have shown that just 2% of older adults meet the recommended levels of physical activity. These inadequate physical activity levels result in a reduction in aerobic capacity, strength, and balance, leading to increased risk for falls, frailty, and functional decline, necessitating the need for physiotherapy.

Physiotherapists are uniquely trained to prescribe and modify physical activity based on the needs of the older adult. Through individualized plans, physiotherapists can empower older adults to complete appropriately dosed physical activity that will prevent functional decline and reduce the risk of falls, frailty, and dependence on others. As movement experts, physiotherapists are the pharmacists of physical activity. Just as a pharmacist would adjust the dosage in a prescription for a person based on sex, body weight, comorbid conditions and medication interactions, physiotherapists are uniquely positioned to do the same based on comorbidities, as well as the person’s needs (impairments and movement assessments), goals, and preferred modes of physical activity (preferences).

Presently, older adult communities are not fully leveraging physiotherapists and their capacity to address the significant public health concern of inadequate physical activity and its subsequent negative consequences. This manuscript presents four key issues linked to inadequate physical activity among older adults and proposes the implementation of four resolutions that physiotherapists can address these problems. See Table 1.

## 2. Four Key Issues, Resolutions, and Implementation Strategies

### 2.1. Key Issue #1: Exercise Is Often Misunderstood as Being Synonymous with Physical Activity

This perception can be a barrier to older adults who have the potential to engage in physical activity for health or recreational purposes but have no interest in exercise [11]. Some older adults perceive traditional medicine to be an easier, quicker, and more effective fix for disease management with minimal side effects, and a more sophisticated way to target a specific disease. For these reasons, many older adults prefer medical management over lifestyle management. Physical activity includes a number of sub-domains such as occupational activities, leisure time, yardwork, transportation, domestic activities, and exercise [5]. Exercise is distinct from other forms of physical activity in that it is structured, repeated, and for the singular goal to improve fitness.

#### Implementing Resolution #1: Shed Light on Physical Activity and Inactivity

Can physical activity improve fitness? Almost all of us in healthcare and wellness pause at this question, when the answer should be a resounding and immediate “yes!”. Exercise can be a very polarizing term as society offers multiple examples of physical, economical, and emotional barriers to exercise. The public often hears sensationalized stories or long held cultural tales. For example, there are various news stories reporting sad cases of sudden deaths in triathlon athletes, increasing arthritis rates that result from running, worn-out joints from CrossFit training, and becoming too bulky from weightlifting. These stories serve as a stiff-arm keeping exercise at bay.

Instead of these stories, the public would benefit from debunking the myth that structured exercise is the only activity that results in better health. Physical activity—movement “my way”—can be equally healthy, require less devoted/novel time, and feel more engaging. Physical activity can feel familiar and within reach, soon becoming a habit when it can be placed inside a person’s everyday movements. An example of healthy physical activity might include gardening or walking a dog. If people do not recognize these movements as contributing to their wellness, then they may feel as though another day went by without enough time to exercise. Redefining physical activity as a health and wellness tool may be effective for increasing the adoption of more movement. For those who feel exercise is not safe, fun, or possible, physical activity can be a very effective strategy. Specifically, physiotherapists should recommend physical activity for disease management and recovery, and as an investment for both brain and body health.

Healthcare providers can use the Age-Friendly Healthcare model to help older adults identify what matters most while creating meaningful activities to meet or exceed the recommended levels of physical activity [12,13]. This model encourages person-centered care through an emphasis on the 4 Ms: (1) What Matters: align healthcare with older adults’ health outcome goals and care preferences; (2) Mentation: prevent, identify, treat, and manage dementia, depression, and delirium; (3) Mobility: optimize physical function and safety; and (4) Medication: minimize the use of inappropriate medications and polypharmacy [13].

Individuals who have difficulty determining what matters to them can utilize self-report assessments for guidance. In the assessment developed and validated by Moye et al., there are four primary domains: (1) functioning (i.e., taking care of self, thinking clearly, walking/moving around by self, and living at home); (2) enjoying life (i.e., participating in hobbies, religious/spiritual life, and physical/sexual intimacy); (3) connecting (i.e., having relationships with family and friends, avoiding being a burden to others, and considering the needs/interest of family); and (4) managing health (i.e., avoiding discomfort, influence of religion/spirituality, and quality vs. length of life) [14]. Secondary to these domains being impacted by physical activity or impacting physical activity, gaining insight about what matters can be used to develop physical activity goals and action plans. These goals and action plans are developed by the patient with guidance from a physiotherapist. A physiotherapist’s role is to empower patients to push themselves to the highest level of intensity while also promoting high self-efficacy. A physical activity goal developed by the patient could be to participate in daily yardwork or gardening lasting 30 min. An action plan developed by the patient could be to complete 30 min of yardwork or gardening every day after breakfast. This will include squatting to pick up items from the ground, walking while carrying gardening tools, and reaching for items outside base of support.

People with persistent conditions may be more successful in the mitigation and management (if not resolution) of their conditions when playing an active role in making decisions about their healthcare—i.e., having autonomy. World populations have an incomplete understanding regarding the benefits of physical activity. More education, however, is not always the primary solution. Education may be the proverbial big stick in healthcare and wellness, which is swung too fast, too often, and too wildly with not enough impact. Populations at large do not need more medicalese or more guilt about moving more. People do not want to read another article filled with loss aversion and coercive comments about the rates of falls, disease, and death in underactive people as compared to their more active counterparts. The knowledge deficit barrier is not bridged with more education; instead, it is bridged with more opportunities and choice. When people have options *and* autonomy, they create their own set of beliefs about what makes sense and what might work for them. They have the autonomy to make choices. When people choose options for themselves (a route, a movie, an exercise plan, a diet, or a relationship), they are more likely to stay committed while tolerating a few ups and downs along the way. In addition, choice means belief. Belief provides a positive boost toward effectiveness, whether the intervention is a sham, or one with efficacy. The ABCs (autonomy, belief, and choice) matter immensely and lead to better outcomes, and thereby more intensity and compliance.

The WHO published recommendations for minimum physical activity for adults in 2010. Over these last 14 years, adoption rates have not moved appreciably. Why? Because education, coercion, mandates, consensus statements, and recommendations do not motivate people. People are motivated to change when the options feel easy, the rationale is logical (obvious), and the results are satisfying. Ask James Clear, the author of the bestselling book about change—*Atomic Habits* (2018) has sold over 14 million copies while basing behavior change on four pillars: easy, obvious, satisfying, and attractive [15].

### 2.2. Key Issue #2: Providers and Older Adults Often Blame Age for Current Health Status and Physical Challenges Confronting Older Adults

Despite recent articles highlighting the negative impacts of ageism, healthcare providers continue to incorporate ageism into their daily practice. For example, when older adults experience knee pain, they frequently blame the symptoms on their age. Providers might push back on this notion, suggesting, “Isn’t your other knee the same age?” While a patient might momentarily consider this, it rarely debunks their bias about age. Similarly, providers can be the source of ageist comments, suggesting to a patient or caregiver, “What do you expect, she’s 82 years old?” or “Falling is just a part of what happens to people when they age.” Relatedly, hip and knee joint replacements are thought by many to be as inevitable as persistent pain, forgetfulness, and falls—they come with aging.

#### Implementing Resolution #2: Avoid Diagnostic Explanations That Weigh Age as the Only Explanation

Ageism is defined as stereotyping (how we think), prejudicing (how we feel), and discriminating (how we act) towards individuals based on their age. Unfortunately, treating individuals differently, secondary to their age, is not a new concept. Ageism has been prevalent for decades, but its ubiquity and insidiousness has led to it being largely unrecognized and unchallenged. Almost all older adults (93%) report experiencing daily ageism [16]. Older adults who experience ageism in healthcare can feel unconcerned or incapable of making independent decisions or fully participating in their own care.

Instead of focusing on age as the primary reason for health conditions, impairments, functional limitations, or participation restrictions, we must identify and resolve the underlying causes. Longitudinal studies have acknowledged that engaging in frequent physical activity is associated with a lower probability of reporting musculoskeletal pain ten years later [17]. Similarly, older adults who perform low, moderate, or high levels of physical activity have a lower likelihood of experiencing musculoskeletal pain compared to those who are sedentary [17]. Thus, inadequate physical activity (leading to losses in strength, kinesthesia, local energy systems, and stiffness), and not age, is often the leading factor resulting in various painful conditions. Similarly, the risk of functional limitations can be decreased by approximately 20% if an older adult increases physical activity from being sedentary to “somewhat active”, and even more if moderately active [18].

One way to combat ageism is to ensure ageist terms, such as those highlighted in Box 1, are avoided. Identifying appropriate terminology starts with the ability to differentiate between younger and older adults. According to the United Nations and the WHO, the age at which one is considered an older adult is 60 years [19]. In contrast, Medicare and the National Institute on Aging classify an individual as an older adult at 65 years [20]. Other age cut-offs used by organizations include 50 and 55. Additionally, the World Economic Forum and highly respected demographers define older adults based on their prospective age, meaning the average number of years left to live. Based on prospective age analyses, individuals are classified as older adults when they have 15 years or less to live [21]. Positive and negative factors contributing to prospective age include health status, disability, income, living arrangement, social support, etc. This classification method highlights the uniqueness of each person’s aging experience and why grouping all older people together as a cohort creates confusion and disparities in healthcare.

Box 1Ageist Terminology.
AgedAncientBed BlockerBiddyCrockDementedElderlyFoggyFossilSilver TsunamiHagOld ManOne Foot in the GraveOver the HillSenileSeniorBoomerDemented


The term *geriatrics* was coined by Ignatz Leo Nascher, who in 1909 presented the following explanation of the term: “Geriatrics, from *geras*, old age, and *iatrikos*, relating to the physician, is a term I would suggest as an addition to our vocabulary, to cover the same field in old age that is covered by the term pediatrics in childhood, to emphasize the necessity of considering senility and its disease apart from maturity and to assign it a separate place in medicine” [22]. However, *geriatrics* did not become a specialty within the medical field until 1988 when a subsequent geriatrics model of care was initially developed in the 1990s–early 2000s. Today, the term geriatrics focuses on the health and wellness of an aging body. Geriatrics is a team approach to caring for older people and supporting family caregivers, including close friends. As suggested by the American Geriatric Society, healthcare providers should advocate to go beyond treating someone who is classified as just being an older adult. Furthermore, healthcare providers should stop using age as an excuse or reason for preventable or modifiable health conditions.

### 2.3. Key Issue #3: Physiotherapy Is Initiated Only When People Experience Injury, Illness, or Inactivity

In many countries, physiotherapy is a covered benefit only for injury, illness, or inactivity, but not wellness or preventative care. Older adults would benefit from participating in community screening events and receiving preventative care from physiotherapists to reduce the risk of frailty or falls.

#### Implementing Resolution #3: Promote the Financial and Physical Benefits of Primary Prevention Models

The American Dental Association has illustrated the power and capability of conducting preventative screenings that lead to improved dental health. This is associated with an improved ability to eat and to communicate, two important functional tasks associated with physical and mental health. Within physiotherapy, an example of how the dental model could be utilized is highlighted by the WAMI-3 model from Billek-Sawhney et al. [23]. This insightful model applies the controllable nature of the origins of frailty from three different perspectives: illness, injury, and inactivity [23]. Each of these three Is (singularly or in combination) can lead to frailty [23]. They share the commonality of primary prevention through physical activity.

Numerous national and international organizations, including the American Physical Therapy Association, American College of Sports Medicine, and WHO call for annual physical activity assessments and fall prevention screenings on every patient or client. There are several tools available to identify inadequate physical activity and the consequences of inadequate physical activity, such as falls and frailty. If these tools are routinely integrated into healthcare and completed in community settings, then healthcare providers would have the power to identify and address sedentary lifestyles, or reduction in activity, prior to adverse consequences occurring. A simple way of identifying inadequate physical activity is using the Physical Activity Vital Sign (PAVS), a two item self-report questionnaire reported to be reliable and valid [24]. Furthermore, the tool has recently been modified to align with global physical activity recommendations, and includes an item on the strength and balance-based physical activities [24,25,26].

Physiotherapists have successfully integrated annual fitness screens into the clinical setting. For example, Puthoff et al. identified that the Adult Functional Independence Test (A-Fit) is a feasible screening tool for physiotherapists to implement in the clinical setting. It also has a high value for identifying aging adults with declines in physical fitness and those who would benefit from further interventions [27]. Another study evaluated the impact of physiotherapy led statewide community-based fall risk screening and identified that within thirty days, 50.0% of screened older adults adopted at least one recommended fall risk-reducing behavior, and by month five, this increased to 64.9% [28]. Furthermore, among the older adults who performed an annual fitness exam completed by a physiotherapist, each older adult had an average number of five deficits, with the range being 0–11, and 50% illustrated impaired strength as defined by hand grip dynamometers and sit-to-stand tests [27]. The older adults were interviewed after the annual fitness exam. Fifty-eight percent indicated they would repeat the annual fitness exam annually, and 42% would pay for this service, with some stating they would pay $100 [27].

### 2.4. Key Issue #4: Medical Diagnoses Determine Function and Quality of Life

Many patients, non-medical people, and some providers succumb to the notion that once a medical condition or diagnosis is determined, it cannot be adequately managed or resolved with health-promoting behaviors, such as physical activity.

#### Implementing Resolution #4: Advance Physiotherapists as Primary Healthcare Providers

Within the United States, there has been a push to include physiotherapy as a part of primary care [29]. Integrating physiotherapists into the primary care team has shown to be effective in healthcare cost savings, reduced wait times, increased utilization of physiotherapy, reduced unnecessary imaging, and reduced opioid prescriptions [29]. As is demonstrated in the military and many other settings, physiotherapists can contribute to first line evaluation of common older adult conditions such as musculoskeletal disorders, vertigo/vestibular concerns, balance deficits and falls, cardiovascular and pulmonary issues, pelvic health, and neurological issues. With this initial contact and evaluation as a primary healthcare provider, physiotherapists can refer older adults to specialized healthcare providers as needed, initiate appropriate physical activity and movement strategies, and monitor patient response to intervention.

With direct access permissions across all 50 states in the United States, and across many countries in the world, physiotherapists are well positioned to be frontline providers for a multitude of reasons—most notably in the management of older adults. The capacity to provide screening in musculoskeletal, neurologic, cardiovascular, genitourinary, cognition and communication, and integumentary matters has been established. Physiotherapists can contribute to first line evaluation and the management of common older adult conditions such as musculoskeletal disorders, vertigo/vestibular concerns, balance deficits and falls, cardiovascular and pulmonary issues, pelvic health, and neurological issues.

Further considerations that support physiotherapists as primary healthcare providers include the increasing evidence that they are effective team members in the emergency department, in capitated healthcare for workforce management, in medical missions, as well as in the standard of practice in some countries. Specifically for geriatric medicine, the considerations and arguments for physiotherapists as primary care providers become even stronger when we think about the primary impairments (and associated functional limitations and participation restrictions) that compromise older adults in almost every country. Physiotherapists are well positioned to screen, examine, diagnose, and manage the frequent impairments experienced by the older adult population, such as sarcopenia, reduced VO_2_ max, fall risk, and chronic pain. This powerful list of four impairments may have a greater negative functional impact and incidence/distribution or commonality across older adults than any other combination. They all fall directly within the scope and capabilities of physiotherapists.

## 3. The Case for Physical Activity: Discussion and Future Opportunities

We have highlighted how incorporating physical activity and primary prevention as a standard of physiotherapy care is crucial for addressing the root causes of the most impactful age-related impairments. The dental model provides a valuable framework, illustrating how preventative screenings can lead to improvements in overall physical and mental health. By adopting a similar approach, physiotherapy can leverage the Age-Friendly Healthcare model and WAMI-3 model to prevent falls and frailty through illness, injury, and inactivity prevention. These models can be used in physiotherapy clinics as illustrated by the above example of the A-Fit, or in the community as illustrated by the above example of the fall prevention community screening. The value of utilizing preventative screenings, particularly related to inadequate physical activity, is further supported through organizations such as the American Physical Therapy Association, the American College of Sports Medicine, and the WHO. Furthermore, all organizations support the elimination of ageism.

A simple physical activity assessment that can be incorporated for all patients and clients, including those 65 years and older, as noted above, is the PAVS. The PAVS includes two items about aerobic activity and one item about strengthening activities. Currently there is a push to also validate a balance item for older adults. By identifying inadequate physical activity levels early, healthcare providers can intervene to eliminate or minimize adverse effects, such as falls and frailty. Once individuals are identified as being inadequately active (i.e., not meeting the WHO Physical Activity guidelines), physiotherapists must incorporate a solution into the plan of care. Examples within the clinic include: (1) using behavior change techniques such as behavioral economics and brief action planning; (2) using specific physical activity prescriptions including Vigorous Intermittent Lifestyle Physical Activity (VILPA); and (3) recommending community programs or resources (i.e., personal trainers, health coaches, etc.). Prior to reading the next paragraphs that focus on these specific examples, please think about the questions and suggestions posed in Box 2.

Box 2Meaningful Questions to Guide Older Adult Physical Activity.
What form of physical activity is the older adult already familiar with?This might include work, sport, or avocational historyWhat form of physical activity does the older adult have easy access to? This might include a nearby trail, court, gym, or home equipmentWhat can we measure (and what is important to them to measure) regarding the older adult’s current function or health profile?These profiles could include a walk test, strength test, steps/day, resting heart rate, or perceived rate of exertionWhat event, goal, or activity motivates the older adult? This could include serving in a caregiving role (spouse/grandchildren), an upcoming vacation, competitive race, health profile, or age-based norm


Friction is a powerful force inhibiting new habits. It is common for people to feel that they do not have time to exercise. A new habit of physical activity may feel more engaging when the movement is familiar, easily measured or accessible (convenient). All these tools (measurement, familiarity, and nudge/convenience) can help to eliminate or overcome friction.

Older adults are more likely to continue a physical activity program which feels meaningful, familiar, convenient, and effective and for some, includes frequent checkpoints of success. As with most behavior changes, starting is the critical first step. Thereafter, a world in motion tends to stay in motion. After one has started a new activity (before it becomes a habit), the stickiness that leads to habit formation can include social connection, gamified wins, external reinforcement (compliments from others), as well as internal sensations (dopamine, serotonin, endogenous opiates). To ensure that the components of the physical activity program are meaningful, they should revolve around and be determined by the older adult.

An example of this is **Brief Action Planning**, a structured motivational interviewing tool that empowers individuals to set their own physical activity goals and action plan. We provide readers with Appendix A which consists of a worksheet that a patient can complete with a physiotherapist or between physiotherapy sessions. In general, Brief Action Planning starts with the physiotherapist asking older adults the following question: “Related to your physical activity, is there anything you would like to do between today and our next session?” If they say yes, a physiotherapist will then ask what they would like to do. This is followed by asking them “Would you like to make a specific plan about that?” An example of a goal and an action plan are: (1) **Goal:** “I will walk from the car to the football field, a distance of 500 m, requiring the ability to walk on grass and a gravel foot path;” and (2) **Plan:** “Every Monday, Wednesday, and Friday morning, I will go to the local park and walk five repetitions of 100 m with a 3-min sitting rest break on the bench every 100 m. Speed of walking will be based on my perceived effort with a target of having difficulty carrying on a conversation.” After a plan is developed, Brief Action Planning recommends identifying the patient’s level of confidence by asking “On a scale 0–10, about how confident do you feel about carrying out your plan?” Those who identify their confidence as <7 are asked to modify the plan to increase their confidence and likelihood of accomplishing their plan. By accomplishing their plan, patients gain a mastery of experience, an essential component of self-efficacy and participation in physical activity.

One way to prescribe physical activity that incorporates intuitive forms of exercise snacking is known as VILPA, or Vigorous Intermittent Lifestyle Physical Activity. As described most frequently by the prolific author Emmanuel Stamatakis [30] refers to brief bouts of vigorous intensity physical activity performed as part of daily living. This might include walking more briskly for 30 s bursts when you are already committed to taking your dog for a walk, or your grandchild out in a stroller. VILPA could include an intentional hustle up the stairs (using a railing as needed), when you are headed up for that left-behind accessory. VILPA can even include a targeted effort to put dishes or groceries away. Moving briefly, with intention and intensity, is healthy. VILPA can make these opportunities feel obvious, salient, and easy. The science on VILPA, again largely associated with Stamatakis, includes findings that just three to four one-minute bouts of VILPA daily is associated with a 40 percent reduction in all-cause mortality and up to a 49 percent reduction in mortality from cardiovascular disease. These have been converted by some to include a 1:4 and as much as a 1:7 ratio of time spent in intense movement to lifespan saved.

The last variable in the equation of influencing physical activity levels is community programs. In the United States and Europe, movements toward this end had excellent momentum in 2018–2020. With the onset of the pandemic, some of these programs faded, and research subsided. Since the pandemic, video-based and remote programs have emerged. Breda and colleagues’ article from 2018 details the European efforts nicely [31]. In the United States, an insurance-based program and excellent resource is Silver Sneakers [32]. Community programs include Enhanced Fitness, etc. Furthermore, both the National Institutes of Health and the National Health services provide online exercise videos that individuals can perform from the comfort of their own home. As noted, emerging industry alternatives include novel programs of remote (live) exercise classes, employing local exercise experts (personal trainers, health coaches, or even a one-time re-evaluation from a local physiotherapist).

Many of us are compelled to move more consistently and with greater intensity through the principle of gamification. As we revisit the examples above of walking a dog or pushing a stroller, when individuals are engaged to track their steps while performing these necessary life events, they can compete against tomorrow’s steps, or their perfect week streak. Recall from Box 2 that one of the options to further engage movement is measurement. Whether it be points, levels, rewards, badges, or streaks—it is all gamification [33]. In each example, the mover receives the serotonin reward from choosing something healthy for themselves and the dopamine reward from the pursuit. It is not necessary, however, that they win. We do not need more steps, flights, or calories than yesterday every day. Near misses, wins, and the pursuit can all deliver our reward chemical, dopamine. The sensations experienced from the release of dopamine become associated with the meaningful movement chosen. These increase engagement with the activity and with movement, and make it more likely that we will begin to identify ourselves as a healthy person. Readers interested in a deeper dive may explore related terms from the field of behavioral economics that can only be mentioned here. Three of the most salient concepts related to physical activity, aging, and gamification include the Near Miss Effect, loss aversion (competing against expected losses), and Protection Motivation Theory. **The only thing we have to fear is frailty.**

As we begin to conclude this paper on physical activity, we must consider how we can leverage physical activity to both reduce the incidence of and recover from the experience of frailty. Fried and colleagues defined the frailty phenotype as a syndrome in their 2001 paper, with the following criteria: a negative energy balance, sarcopenia, along with diminished strength and a low tolerance for exertion [34]. The authors continued, offering five criteria, any three of which can be present to define frailty [34]. These criteria include low grip strength, low energy, slowed waking speed, low physical activity, and/or unintentional weight loss [34]. In a 2011 related paper, Xue defined frailty as “a clinically recognizable state of increased vulnerability resulting from aging-associated decline in reserve and function across multiple physiologic systems such that the ability to cope with everyday or acute stressors is comprised” [35]. This definition represents both how far we have come, and how far we have still to go, with the ageism exhibited in this peer-reviewed publication, which blatantly associates frailty as an age-related syndrome rather than one that can occur independent of age [35]. As we pivot from the notion that frailty comes with aging or is one phenotype/expression of how some people age, we change the image of the condition from one that is uncontrollable, to one that can be prevented, and recovered from.

It is important and plausible to prevent some frailty in physiotherapy. Persons at greatest risk for frailty can be readily identified using the WAMI-3 as well as community screening tools. Physiotherapists can leverage modes of physical activity that feel both purposeful and engaging to the person at risk, in an effort to achieve measurable gains. As Danquah and colleagues suggest in their 2024 article, personalizing a physical activity prescription may be better-received, engage toward greater intensity, and deliver more of a mental health benefit than an exercise-only prescription [36].

## 4. Conclusions

Physiotherapists are the pharmacists of physical activity. Physiotherapists have the training and time to evaluate and prescribe the optimal level and mode of physical activity considering personal preferences, condition, and fitness (capacity). To provide autonomy and foster adoption, the authors propose and science supports the notion that physical activity may feel universally more approachable to all, as compared to the notion of exercise. In this paper we have detailed the physiological, psychological, and functional differences between exercise and the more inclusive umbrella of physical activity. While our beliefs are evolving and reducing what primary and secondary aging can and should be blamed for, we assert that informed proactive decisions and actions better determine our destiny and relationship with health conditions than previously thought. Finally, through this perspective paper, readers can appreciate an expanding role for physiotherapists both pre-disease and in reversing or decelerating health conditions that were once determined to be permanent. These persistent conditions include frailty, cardiometabolic dysfunction, complications from diabetes, and many more. While it was only recently conceptualized that exercise is medicine, we champion a more comprehensive viewpoint that physical activity is medicine, empowering our older adults to positively experience the approachability of consuming and mixing this elixir in the form that suits them best.

## Figures and Tables

**Table 1 healthcare-13-00834-t001:** Four Key Issues Impacting Physical Activity in Aging and Proposed Resolutions.

Key Issue	Proposed Resolution
Exercise is often misunderstood as being synonymous with physical activity.	Shed light on physical activity and inactivity.
2. Providers and older adults often blame age for current health status and physical challenges confronting older adults.	Avoid diagnostic explanations that weigh age as the only explanation.
3. Physiotherapy is initiated only when people experience injury, illness, or inactivity.	Promote the financial and physical benefits of primary prevention models.
4. Medical diagnoses determine function and quality of life.	Advance physiotherapists as primary healthcare providers.

## Data Availability

Not applicable as this article has no data sets and cites all relevant references.

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
