# Peer review of "The Pharmacists of Physical Activity: Physiotherapists Empowering Older Adults’ Autonomy in the Self-Management of Aging with and Without Persistent Conditions"

_healthcare, 2025, doi:10.3390/healthcare13070834_

Round 1

Reviewer 1 Report

Comments and Suggestions for Authors

Thank you for allowing me the opportunity to review this manuscript. I agree whole-heartedly with the sentiment of the topic, the title, and the issues. Thank you for caring for older adults in this manner.

Summary Feedback: I do have significant concerns about the presentation of the issues at hand. The paper lacks structure and organization. It presents the issues, resolutions and implementation with very limited structure, consistency, and very little supporting evidence. Where evidence is provided it is weak, sometimes overstated and might not be the best source for the argument being made. The discussion does not review what has been presented but adds new content and there is no summary or conclusion. The title is misleading as it suggests precision in prescription, action by physiotherapists, and specific attention to those with and without conditions. I would point out that the article lacks precision in prescription and does not attend to the remaining claims in the title. I agree that exercise is medicine, and that exercise is the most potent pill available, but simply having a pill does not make one a pharmacist. The role of the physiotherapist dispensing exercise as it would align with a pharmacist dispensing medication is not fulfilled in the article. 

My recommendation is to improve the outline of the paper with greater research support and clarity for action items. Specific suggestions can be found in the comments below.

ABSTRACT

  • Physiotherapists do not reverse risks in older patients. They can reduce risk by a great magnitude, but to make an older adult immune to risks such as falls is overstated.
  • Move the definition of primary and secondary aging to the body of the paper or include it in both rather than count on the definition to be read and/or retained by the reader of the abstract once they start the paper.
  • Quotations and parentheses in the abstract seem to detract from the content. These statements can be written without denotations. Their use may appear to diminish their credibility. For instance, “physical activity is medicine” is not a stretch or untruth but a statement that physiotherapists abide by. Without quotations you can state: Physiotherapists function as pharmacists of physical activity, well-positioned to prescribe…
  • I would encourage the authors to avoid first-person and adopt third person in the abstract and throughout the paper.
    • Thus remove, “in this paper we” and consider, “This paper provides guidance from the perspective of the physiotherapist on exercise prescription most optimal and palatable for an older adult population.”

INTRODUCTION

  • Include a definition of primary and secondary aging.
  • I would continue to recommend removing the quotes that are not critical to the understanding of content such as “exercise” as these seem distracting and, again, detract from the importance of the message. I will not comment on them further. However, when quoting and citing things like the exact WHO definition of physical activity it may be appropriate to use quotations as you did for Dr. Butler.
  • [page 2] It seems you miss an opportunity to start the conversation on ageism here. You develop it later, but it would be appropriate here.
  • [page 2 last paragraph before 2. The Four Key Issues]
    • I think you can just stick to physiotherapists and not define physical therapists.
    • consider a different word than “reverse” when addressing the risk of falling. While it sounds fun and we know we can really change the risk, the term reverse is not accurate. Reverse seems to be overzealous, scientifically inaccurate, and thus probably less persuasive to the naysayer. I agree with promoting what we can do as physios but we should not overstate our potential as we work to convince others of our merit.
    • Remove “our” and rephrase to, “This manuscript presents four…”
  • [2. The Four Key Issues and Resolutions]
    • [first paragraph] the list is the issues, not the resolutions. The organization from here forward is a bit confusing.
      • Suggestions for improving organization.
        • Create a table with proposed issues and a resolution statement.
        • Then move through each resolution with supporting evidence and implementation strategies rather than presenting each issue with resolution and then presenting the implementation of each. There was redundancy and with the summary of resolutions and the difference between the resolution and the implementation were not clear.

**Note***The feedback below is organized by issue/resolution topic and does not stay in written order, but does note the section or page being reviewed in brackets before the comment.

  • [2.1, page 2] The issue has not been supported with citations. Some supporting evidence could be cited and would support a stronger argument.
  • [2.1] You present a model with some definition but if you added an example or some data to support how this has been or could be successful for meeting or exceeding physical activity goals it would be more powerful. Further, stating that the issue of exercise as synonymous with physical activity and then solving the issue with meaningful physical activity leaves the reader potentially confused. Is it synonymous or should the reader be compelled toward promoting purposeful exercise and not just physical activity? The outcome of meeting physical activity with meaningful activity seems to blur the line you present as the issue.
  • [Implementing resolution 1, page 4]
    • On page 4 you outline the 4M healthcare model where it seems that just the first M relates closely to your resolution or you don’t expand on how the other M’s really do. I would encourage a look at motivation literature, qualitative studies that have looked at how older adults perceive exercise vs. activity etc. 
    • In this implementation it is not clear if the action plan suggested directly states that the gardening must include squatting, reaching out of BOS etc. The implementation of increased activity that includes gardening could omit much of this for some individuals (i.e. one who sits on a stool to weed and gets their spouse to reach things for them). These individuals might erroneously count gardening, as encouraged here, as physical activity. Self-reported exercise and perceptions of physical activity are known to be flawed. We do know in terms of dosing physical activity that it needs to exceed what the baseline is and that the dose really matters (as stated in your title). How this is achieved with physical activity models is not made clear in this article and might minimize intentional physiotherapy dosing of exercise or activity.
  • [2.2] I agree that this is a HUGE issue in healthcare and represents an ageist mindset in healthcare providers. I would suggest some citations to support this as occurring rather than just anecdotal quotes.
  • [2.2] The resolution, “Eliminate ageism from all sources” is not an actionable or realistic resolution and does not fit with the summary below it. Is the resolution: “Avoid diagnostic explanations that weigh age as the only explanation.” Add citations. For instance, we know that we can reduce supposed age-related conditions such as sarcopenia with purposeful exercise.
  • [page 5 Resolution 2]
    • Consider adding a citation for ageist terminology. There are many sources that support this.
  • [page 6 Resolution 2]
    • Beyond avoiding ageist terminology could more be said regarding the known inconsistency of treating a patient based on age? It seems there is more to implement here. In my opinion, a key phrase that is problematic and missing from the box is, “For your age.”
    • Finally, there ARE age-related changes in all adults that cannot be overcome. These must be acknowledged and addressed by skilled physiotherapists as older adults can be amazing at compensating, reducing the impact of age-related change etc. Some things do change with age.
  • [2.3] This is another lofty and unachievable resolution. While it is the ideal, it lacks inspiration for being unattainable. Consider, “Promote the financial and physical benefits of primary prevention models.” Or something more tangible and inspiring.
    • I would recommend summarizing the benefits into 1) financial 2) reduction of risk 3) compliance or the like. You’re focusing on 2 studies in your justification which seem to provide a weak argument for the model as they do not show actual cost savings nor do they demonstrate a reduction in actual falls. Evidence that supports primary prevention with these metrics would be more convincing.
  • [Implementation of 3 on page 6]
    • Any evidence to support implementation of primary prevention with physical therapy should be clearly outlined and cited here. I would start with the Puthoff et al. study as it is the contemporary model of this. The 3 I’s is proposed but stops short of demonstrating efficacy. The call from ACSM is convincing but not cited. The PAVS seems to not require physiotherapists and while the authors state it is valid and reliable the citations provided do not support this. Even a small study showing benefits in areas such as financial, risk, health, exercise compliance would be compelling.
  • [2.4] The issue stated here is not consistent with that in the summary list at the start.
    • Add citations to support the issue.
    • Rephrase the resolution to a consistent action. This is a statement. Change all to actions or all to statements.
    • The citation (14) is another perspective paper. The support of the benefits listed are not from a meta-analysis or systematic review and could be bias. It might be better sourced from the primary sources within reference 14.
  • [implementation of 4, page 7]
    • There are no supporting references, any arguments made here would be enhanced with objective numbers and supporting references.
  • [4. The Case for Physical Activity page 7]
    • The summary encourages the use of the WAMI-3. There is no clear summary about how this model might be implemented and no evidence cited to support its efficacy. I recommend not using it unless both are included.
    • The suggestion to implement PAVS is a self-report measure that does not require a physio to complete. I’m not sure why this non-skilled tool is something the authors are leaning into?
    • Box 2 seems arbitrary. I assume it would be made to be more visually appealing, but with tools that exist to gauge some of these constructs the list lacks support from the literature. Further, it is not summarized in a manner that will facilitate use. What are the key constructs here? 1. Mode 2. Mechanism 3. Measurement 4. Motivation? A reader may be more likely to remember such a list that aligns with the constructs.
    • The Brief Action Planning and VILPA both seem like fine ways to objectively seek more activity. If generalized activity is the goal, what is the role of the physical therapist? The title of this article approaches the task as a Pharmacologist but there seems to be no attempt to dose exercise here. The abundant encouragement of general activity/exercise is a recipe for many older adults to get hurt and quit as they do not have the information, dosing, guidance needed to engage purposefully and safely in exercise/activity.
    • What is the role of the physical therapist in the community programming? A recheck?
    • The definition of frailty is brought up in the last paragraph. Frailty has been discussed already with the WAMI-3. Where is the paper going here?
    • There is no clear conclusion and rather, a criticism of the definition of frailty at the end of this piece. A summary and call to action that is clear and organized would be warranted.

Reviewer 2 Report

Comments and Suggestions for Authors

After analyzing the study, here are my considerations to the authors.

Abstract
1. I suggest that this section be corrected, as it lacks a clear objective, presentation of methods, as well as a lack of presentation of results and conclusions.
2. In general, the current Abstract section fully presents the state of the art.

Introduction
1. Although the text is well written and aims to present something interesting, it stops at presenting a series of facts supported by 7 references;
2. The following statement requires theoretical basis: [(...) physiotherapists can empower older adults to complete
appropriately dosed physical activity that will prevent functional decline and reverse the risk of falls, frailty, and dependence on others. As movement experts, physiotherapists are the “pharmacists of physical activity”].
3. In addition, this aforementioned theoretical basis would be important so that there is no conflict between the areas of Physiotherapy and Physical Education. In general, the literature attributes the promotion of "physical activity" to Physical Education professionals. Therefore, it is necessary to clarify this topic;

4. Before or after, the authors presented the following sentence "In our manuscript we present four key issues linked to inadequate physical activity among older adults and four potential resolutions for how physiotherapists can address these problems"; it is essential that the introduction has content that supports these 4 points.

Methods - The study does not present a defined method. Based on the content, a qualitative line is identified: review.

Results - They are adequate. However, they lack a justification methodological design [Introduction].

Discussion - This section occurs together with the results. However, this is not clear in the study methodology. * Lack of a section on Limitations, Strengths of the study, Future research.

Conclusion - There is no such section

Round 2

Reviewer 2 Report

Comments and Suggestions for Authors

Dear authors, after rereading the manuscript, I consider that the text is in a better shape compared to the previous version. Therefore, it is ready to be published.
Sincerely
Reviewer